# Study on Multi-Model Soft Sensor Modeling Method and Its Model Optimization for the Fermentation Process of *Pichia pastoris*

**DOI:** 10.3390/s21227635

**Published:** 2021-11-17

**Authors:** Bo Wang, Xingyu Wang, Mengyi He, Xianglin Zhu

**Affiliations:** Key Laboratory of Agricultural Measurement and Control Technology and Equipment for Mechanical Industrial Facilities, School of Electrical and Information Engineering, Jiangsu University, Zhenjiang 212013, China; wangbo@ujs.edu.cn (B.W.); hmy1145755423@163.com (M.H.); zxl4390@126.com (X.Z.)

**Keywords:** soft sensor, piecewise affine, improved compression factor, particle swarm optimization, *Pichia pastoris*

## Abstract

The problems that the key biomass variables in *Pichia pastoris* fermentation process are difficult measure in real time; this paper mainly proposes a multi-model soft sensor modeling method based on the piecewise affine (PWA) modeling method, which is optimized by particle swarm optimization (PSO) with an improved compression factor (ICF). Firstly, the false nearest neighbor method was used to determine the order of the PWA model. Secondly, the ICF-PSO algorithm was proposed to cooperatively optimize the number of PWA models and the parameters of each local model. Finally, a least squares support vector machine was adopted to determine the scope of action of each local model. Simulation results show that the proposed ICF-PSO-PWA multi-model soft sensor modeling method accurately approximated the nonlinear features of *Pichia pastoris* fermentation, and the model prediction accuracy is improved by 4.4884% compared with the weighted least squares vector regression model optimized by PSO.

## 1. Introduction

An expression system is a molecular biology technique that uses model organisms such as bacteria, yeast, animal cells or plant cells to express exogenous gene proteins. In short, the expression process is the synthesis of proteins under the guidance of genes. The *Pichia pastoris* expression system is a technically mature eukaryotic expression system, and one of the most successful foreign protein expression systems [1]. The expression system of *Pichia pastoris* has been developed for more than 30 years, and so far, nearly 1000 heterologous proteins have been successfully expressed by this system, which has a promising development [2]. In addition, the low cost but high yield of the *Pichia pastoris* expression system makes it economically valuable. *Pichia pastoris* is also a facultative anaerobe, which is very easy to carry out genetic manipulation and cultivation on [3]. The *Pichia pastoris* expression system has obvious advantages in the processing, external separation, post-translational modification, and glycosylation modification of expression products [4]. Nowadays, many regions are being plagued by the Coronavirus (COVID-19). In order to fight COVID-19, the recombinant expression of proteinase K in *Pichia pastoris* has attracted widespread attention [5,6]. Protease K has drawn attention for its following capabilities: cracking the COVID-19 virus to release nucleic acid, eliminating ribonuclease (RNase) to prevent ribonucleic acid (RNA) degradation, and inactivating the COVID-19 virus to denature the virus protein [7]. Therefore, protease K becomes an important component of nucleic acid detection kit for sample pretreatment, and plays a significant role in COVID-19 detection [8]. However, the fermentation process of recombinant expression of protease K by *Pichia pastoris* is nonlinear, coupled, uncertain, and time-varying [9]. There are few online detection instruments for the key biomass variables of the *Pichia pastoris* fermentation process (such as cell concentration, protease K concentration, etc.) [10] As for the offline assay and analysis methods, the long interval of data acquisition drags down the real-time performance, and adds to the risk of flora pollution. Moreover, the assay results are often inaccurate, due to the large errors of human operation [11]. With the development of the times, biosensors that can detect biomass on-line in real time emerge as the times require. Biosensors have the advantages of high sensitivity and a fast analysis speed; however, a highly automated, miniaturized and integrated biosensor is often very expensive [12].

The above problems give rise to the soft sensor technology. The basic idea of this technology is to model the relationship between target and auxiliary variables, and thus estimate the target variables indirectly, without causing any pollution to flora. The Soft sensor technology is powerful and cost-effective, and it can work in harmony with other software and hardware, which means that it is highly compatible [13]. However, in most cases, it is difficult for a single global soft sensor model to adequately and quickly characterize the process of a complex object. Furthermore, a single global soft sensor model is usually complex in structure and requires a lot of time to solve it. Wang and Han [14] proposed a single soft sensor model based on a recurrent wavelet neural network and Gaussian process regression methods for online prediction of whether Chlortetracycline fermentation broth is contaminated by non-target bacteria. Although the experimental results show that the proposed soft sensor model can be used to predict the occurrence of pollution during the Chlortetracycline fermentation process, and the effectiveness of the modeling method is verified based on field data, the Chlortetracycline fermentation process is a very complex and non-linear process, and most microbial fermentation processes including Chlortetracycline can be divided into four stages: adjustment, logarithmic growth, stabilization and decay phases. During these four phases, the process characteristics of Chlortetracycline are completely different, especially during the exponential growth phase, when the fermentation process is most intense. Therefore, it is impractical to try to describe the complete fermentation process by a single soft sensor model. Although the soft sensor model proposed in this paper showed good prediction results for the Chlortetracycline fermentation process, the model has limitations and cannot be extended to soft sensor modelling of other microbial fermentation processes, i.e., it is not generalizable.

Therefore, in order to address the drawbacks of a single soft sensor model, multi-model soft sensor modeling strategies have attracted a great deal of scholarly attention [15,16,17]. Scholars have found that using the divide and conquer modeling strategy, building a multi-local soft sensor model can greatly simplify the model structure, save calculation time, and fully mine the information of each sample, hence multi-local soft sensor modeling methods are becoming an inevitable trend in the development of soft sensor technology in the future [18]. As a result, it is highly practical to build a multi-local soft sensor model that supports the real-time online measurement of key biomass variables in *Pichia pastoris* fermentation, as well as the efficient (protease K yield) and high-quality generation of products (protease K quality).

Following the above ideas, Zhang and Cai [19] presented a multi-model soft sensor modeling method based on the improved locally weighted partial least squares (LWPLS), and adopted the soft sensor model to predict the key biomass variables in the fermentation process of alkaline protease. Their approach copes well with the multi-stage and large lag features of microbial fermentation, and realizes good prediction accuracy. However, it is difficult to identify the structural and local parameters of the extremely complex model, and it is time-consuming to complete the heavy calculations. Hence, their approach is not applicable to a wide range of fermentation processes.

In order to avoid the problems mentioned above, this paper will apply a multi-model modeling method called piecewise affine (PWA), which is also called piecewise linear (PWL). Based on the idea of multi-model “decomposition-synthesis”, the PWA/PWL multi-model divides the sample space of the system into a finite number of sub-intervals, and each sub-interval is described by a PWA/PWL local model, which transforms the problem of solving a complex nonlinear system into the problem of solving several linear systems [20]. The advantages of the PWA multi-model are that the model structure is very simple; after all, the structure of linear systems is far less complex than that of nonlinear systems, and PWA multi-model is less time consuming than other multi-models because the time to solve a linear system is much less than that to solve a nonlinear system [21,22]. These advantages have made PWA systems a hot research topic in the field of modeling, and they are widely used in many engineering practices. Sun and Wu [23] applied a PWA model to speed regulation in longitudinal dynamics of intelligent vehicles. Muhammad and Michael [24] applied a PWA modeling method to optimal planning of thermal energy systems for microgrids. Sindareh-Esfahani [25] identified a discrete-time PWA model of a wind turbine during Maximum Power Point Tracking region. Mattsson and Zachariah [26] applied the PWA modeling method to the identification of discrete-time nonlinear dynamic models of cascade tanks. All these cases proved the effectiveness and feasibility of the PWA modeling method. Therefore, in this paper, a PWA modeling method was applied to soft sensor modeling of key variables in the fermentation process of *Pichia pastoris*.

In order to improve the prediction accuracy of the PWA multi-model, two model parameters must be optimized, i.e., the number of local models and the parameters of each local model. Too many local models will lead to complex models with a large computational effort and huge dimensionality of local model parameters while too few local models will lead to too simple models with poor prediction performance. In recent years, the research on intelligent optimization algorithms and its applications in model parameter optimization has been very active and has achieved encouraging results [27,28,29]. Inspired by the foraging behavior of birds, a PSO algorithm as an intelligent swarm optimization algorithm has been successfully applied to the optimization of model parameters [30,31,32]. The PSO algorithm has no crossover and variational operations, relies on particle velocity to complete the search, and only the optimal particle passes the information to other particles in the iterative evolution, which makes the search speed very fast. The PSO algorithm requires fewer parameters to be adjusted and has a simple structure, which is an effective method to solve practical engineering problems. Many numerical examples show that PSO is better than Differential Evolution (DE), Genetic Algorithm (GA) and Ant Colony Algorithm (ACO) in terms of search efficiency and convergence speed of the global optimal solution [33,34,35]. This is why this algorithm is used in this paper to optimize the number of local models and the parameters of each local model. However, the PSO algorithm also has some drawbacks, such as the lack of dynamic adjustment of the speed, which is easy to fall into local optimum and requires the selection of appropriate parameters for different problems to achieve the optimal results. Therefore, it is essential to improve the PSO algorithm before the PSO algorithm is used to optimize the number of local models and the parameters of each local model. The reason why the PSO algorithm easily falls into local optimum is that the particle velocity cannot be dynamically adjusted, so the particle velocity must be constrained. The most effective way to constrain the particle velocity is to introduce a compression factor into the particle velocity update formula [36]. However, the traditional compression factor has a fixed form and poor self-adaptability, which makes it difficult to achieve the desired constraint effect [37]. Therefore, this paper will study an improved compression factor (ICF) that can make the compression factor change dynamically as the number of PSO iterations increases [38]. Hence, this study eventually uses this ICF-PSO algorithm to optimize the number of local models and the parameters of each local model. The selection of specific parameters and performance analysis of the ICF-PSO algorithm will be analyzed in detail in the following. The complete PWA modeling process requires not only the optimization of the parameters of the PWA multi-model, but also the determination of the order of the PWA multi-model and the determination of dividing surfaces where each sub-model plays a role, all of which will be explained in detail in the following. To verify the feasibility of the ICF-PSO-PWA multi-model, this study obtained real and accurate original sample data through real fermentation experiments. Supported by these sample data, this study demonstrated the feasibility of applying the PWA multi-model to the *Pichia pastoris* fermentation process through MATLAB simulations, and compared the ICF-PSO-PWA multi-model with the weighted least squares support vector regression model optimized by standard PSO algorithm (PSO-WLSSVR). The simulation results showed that the ICF-PSO-PWA multi-model can effectively predict the key biological variables of the *Pichia pastoris* fermentation process online, and it outperformed the PSO-WLSSVR model.

In summary, the purpose of this study can be summarized into two aspects. The primary study focus is to construct and solve a PWA soft sensor multi-model based on the fermentation process of *Pichia pastoris*, and to demonstrate the effectiveness of the PWA soft sensor multi-model through simulations. The secondary study focus is to propose a novel ICF-PSO optimization algorithm to optimize the PWA soft sensor multi-model, which not only serves the primary study focus but also is an innovative point of this paper.

The rest of the paper is structured as follows: Section 2 consists of materials and methods, which explains PWA model basics, determination of the PWA multi-model order, proposed ICF-PSO algorithm, determination of dividing surfaces and experimental work. Section 3 includes results and discussion. The paper is concluded in Section 4.

## 2. Materials and Methods

### 2.1. Structure Diagram of Overall Modeling Algorithm

In order to clarify the components of the prediction algorithm used in this study and their respective functions, this paper first constructs a diagram of the overall modeling algorithm structure here. The functions of each component are then explained in detail in the following sections of this paper. The overall modeling algorithm structure is shown in Figure 1.

### 2.2. Description of PWA Modeling Algorithm 

The existing sample data evidently testifies the nonlinearity of *Pichia pastoris* fermentation. To facilitate the PWA modeling, the *Pichia pastoris* fermentation (Multi-input Multi-output Nonlinear System) was decomposed into several multiple input single output (MISO) discrete time systems, each of which can be described as a piecewise auto-regressive exogenous (PWARX) model [39]:(1)y(k)=f(X(k))+ε(k)
where, y(k)∈R is the system output; ε(k)∈R is the error term.

The PWA mapping f(●) can be defined as:(2)$$f\left(X\left(k\right)\right)=\left\{ \begin{array}{cc} \left[X^{T}\left(K\right),1\right]\theta_{1} & X\in \Omega_{1}\\ \left[X^{T}\left(K\right),1\right]\theta_{2} & X\in \Omega_{2}\\ \vdots & \\ \left[X^{T}\left(K\right),1\right]\theta_{s} & X\in \Omega_{s} \end{array}\right.$$
where, *X*(*k*)∈*R^n^* is the input vector, which consists of the output and input of the system in the past moment:(3)$$X\left(k\right)={\left[y\left(k-1\right),\cdots ,y\left(k-n_{y}\right),u_{\;1}\left(k-p-1\right),\cdots ,u_{\;r}\left(k-p-1\right),\cdots ,u_{\;1}\left(k-p-n_{\;u}\right),\cdots ,u_{\;r}\left(k-p-n_{\;u}\right)\right]}^{\;T}$$
where, *k* is a time series; y∈R is the system output; $u\in R^{r}$ is the system input; *p* is the system delay; *n_y_* and *n_u_* are the orders of the model (*n* = *n_y_* + *n_u_*); *S* is the number of models; $\theta ={\left[\theta_{1},\theta_{2},\cdots ,\theta_{S}\right]}^{T}\in R^{\left(n_{y}+n_{u}r+1\right)\times S}$ is the parameter vector of each linear local model of the PWA system.

The bounded input space *Ω_N_* can be decomposed into *S* closed polyhedral regions ${\left\{\Omega_{i}\right\}}_{i=1}^{S}$. Assuming that these closed polyhedral regions have no overlapping parts except the common boundary:(4)$$\left\{ \begin{array}{l} \overset{S}{\underset{i}{\cup }}\Omega_{i}=\Omega_{N}\subset R^{n}\\ \Omega_{i}\cap \Omega_{j}=\emptyset \;\;\;i\neq j \end{array}\right.$$
where, *Ω_i_* is a closed polyhedron: $\Omega_{i}=\left\{X\in R^{n}|H_{{}_{i}}{\left[X^{T},1\right]}^{T}\geq 0\right\}$, with $H_{i}\in R^{q_{i}}$ which determines the boundary of a closed polyhedron *Ω_i_*, i.e., the dividing surface dividing *Ω _i_* from other polyhedrons (*q_i_* is the number of linear inequalities defining the *i*-th polyhedron region), and ≥ being an inequality:$$\mathrm{If}\;H_{i}=\left[\begin{array}{l} h_{i1}\\ h_{i2} \end{array}\right]h_{i1}\in R^{n+1}\;\;\mathrm{and}\;h_{i2}\in R^{n+1}$$
$$\mathrm{Then}\;H_{i}{\left[X^{T},1\right]}^{T}\geq 0\Leftrightarrow \left\{ \begin{array}{l} h_{i,1}{\left[X^{T},1\right]}^{T}\geq 0\\ h_{i,2}{\left[X^{T},1\right]}^{T}\geq 0 \end{array}\right.$$

Suppose the PWA model is unknown, and the modeling space consists of *N* datasets on the fermentation process:(5)$$Z_{N}={\left\{z\left(k\right)\right\}}_{k=1}^{N}\;\;\mathrm{and}\;z\left(k\right)={\left[X^{T}\left(k\right),y\left(k\right)\right]}^{T}$$
where, *Z_N_*∈*R*^(*n* + 1)*N*^. Then, the identification problem of the PWA aims to reconstruct the PWA mapping to characterize all the features of the original system, that is, to solve the following optimization problem:(6)$$\begin{array}{ll} \min J_{1} & =\min{\sum_{k}\left[y\left(k\right)-\overset{\wedge }{y}\left(k\right)\right]}^{2}\\ & =\min{\sum_{z\left(k\right)\in Z_{N}}\left[y\left(k\right)-\overset{\wedge }{f}\left(X\left(k\right),\theta_{i}\right)\right]}^{2} \end{array}$$
where, $\overset{\wedge }{y}\left(k\right)\in R$ is the output predicted based on the input vector *X*(*k*) at time *k*; $\overset{\wedge }{f}\left(●,●\right)\in R$ is the local PWA model of the current work space:$$\mathrm{Let}\;\delta_{i}=\left\{ \begin{array}{ll} 1 & X\left(k\right)\in \Omega_{i}\\ 0 & X\left(k\right)\notin \Omega_{i}\; \end{array}\right.\Omega_{i}\subset \Omega_{N}$$

Then, Equation (2) can be represented as:(7)$$\overset{\wedge }{f}\left(X\left(k\right),\theta_{i}\right)={\sum_{i=1}^{S}\left[X^{T}\left(k\right),1\right]}^{T}\theta_{i}\delta_{i}\;\mathrm{and}\;X\left(k\right)\in \Omega_{N}$$

Therefore, the solution problem of MISO nonlinear system in each fermentation process can be transformed into a mixed integer quadratic programming (MIQP) problem [40]:(8)$$\begin{array}{l} \underset{n,\theta ,S}{\min}J_{1}=\underset{n,\theta ,S}{\min}{\sum_{i=1}^{N}\left[y\left(k\right)-\overset{\wedge }{f}\left(X\left(k\right),\theta_{i}\right)\right]}^{2}\\ s.t.\overset{\wedge }{\;f}\left(X\left(k\right),\theta_{i}\right)={\sum_{i=1}^{S}\left[X^{T}\left(k\right),1\right]}^{T}\theta_{i}\delta_{i} \end{array}$$
where, $\delta_{i}=\left\{ \begin{array}{ll} 1 & X\left(k\right)\in \Omega_{\;i}\\ 0 & X\left(k\right)\notin \Omega_{\;i}\; \end{array}\right.$, y(k)=f(X(k)), $X\left(k\right)\in \Omega_{N}$ and ${\left[X^{T}\left(k\right),y\left(k\right)\right]}^{T}\in Z_{N}$. It is not hard to see from the objective function *J*_1_ that MIQP only identifies *θ_i_*. However, in order to identify *θ_i_*, several prerequisites are needed, namely, solving for the order *n* of the input vector *X*(*k*) and determining the number of local models *S*. The problem of identifying *n* and *S* is a problem of identifying two hyperparameters, and in this study the optimal *n* and *S* will be solved by different algorithms respectively. When *S* and *n* are known, the objective function *J*_1_ becomes an easier-to-solve MIQP problem containing only one decision variable *θ_i_*. It is only necessary to find the optimal *θ_i_* by the corresponding optimization algorithm for *J*_1_. After all the above parameters are determined, the range *Ω_i_* in which the local model works can be easily determined by just the corresponding classification algorithm.

Without sufficient prior information, however, it is usually difficult to determine the number of models or the parameters of each local model. Therefore, this paper modifies the goal of optimization modeling from maximizing model accuracy by single optimization to maximizing the approximation degree of the model while minimizing the number of local models. That is, the single-objective Problem (6) was transformed into a multi-objective optimization problem:(9)$$\min\left\{J_{1},J_{2}\right\}$$
where, $J_{2}=S$ is an integer representing the number of local models; $J_{1}$ is given by Equation (8), reflecting the approximation of the model. The value of $J_{1}$ is negatively correlated with the approximation degree of the model. Thus, the goal of maximizing the approximation degree of the model is equivalent to minimizing the objective function $J_{1}$.

### 2.3. Determining Model Order

In the absence of sufficient prior information, it is a complex issue to determine the appropriate order of the PWA model. The topological delay embedding theorem [41] suggests that a nonlinear system cannot be completely modeled if the final model has a too-small order; if the model order is too large, the modeling can be complete in theory, but the actual modeling will require heavy computations, and face lots of system noises and a large rounding error [42].

The false nearest neighbor (FNN) method is suitable for determining the embedding dimension in the state space reconstruction of nonlinear time series. The core idea of this method is to iteratively determine the false and real neighboring points of a certain trajectory *x_n_*, while increasing the embedding dimension in each iteration. The iterative process terminates when the number of false neighbors is zero, and outputs the minimum required embedding dimension. From the geometric perspective, the FNN is a very simple and easy approach to determine the embedding dimension [43]. Hence, this paper adopts this method to calculate the order of the PWA multi-model.

Step 1. Determining the false nearest neighbor

Considering the input/output data on fermentation, construct the following *n*-dimensional regression vector:(10)$$X\left(k\right)={\left[y\left(k-1\right),\cdots ,y\left(k-n_{y}\right),u_{\;1}\left(k-p-1\right),\cdots ,u_{\;r}\left(k-p-1\right),\cdots ,u_{\;1}\left(k-p-n_{\;u}\right),\cdots ,u_{\;r}\left(k-p-n_{\;u}\right)\right]}^{\;T}$$

Rewrite the regression vector *X*^(*n*)^(*k*) as:(11)$$x_{p}={\left[x_{p-1},x_{p-2},\cdots ,x_{p-n_{y}},x_{p-\left(n_{y}+1\right)},\cdots ,x_{p-\left(n_{y}+n_{u}\right)},\cdots ,x_{p-\left(n_{y}+n_{u}r\right)}\right]}^{T}$$
(12)$$p=N_{0},N_{0}+1,\cdots ,N$$
where, $N_{0}=n+1$.

Define the distance from *x_i_* to *x_j_* in *R^n^* by $L_{\infty }$ norm:(13)$${\left\|x_{i}-x_{j}\right\|}_{\infty }=\max\left|x_{i+1}-x_{j+1}\right|$$

Let *x_p_* be the nearest point of *x**_η_**_(p)_*:(14)$${\left\|x_{\eta }{}_{\left(p\right)}-x_{p}\right\|}_{\infty }^{\left(n\right)}=\underset{j=N_{0},\cdots ,N;j\neq n}{\min}{\left\|x_{p}-x_{j}\right\|}_{\infty }=\underset{j=N_{0},\cdots ,N;j\neq n}{\min}\underset{\;0\leq l\leq n-1}{\max}\left|x_{p+l}-x_{j+l}\right|$$

Then, *N*_0_ ≤ *η*(*p*) ≤ *N*, and *η*(*p*) is related to *n*, that is, *η*(*p*) is related to *n_y_* and *n_u_*. 

As the embedding dimension changes from *n* to *n* + 1, the distance from *x_i_* to *x_j_* becomes:${\left\|x_{\eta }{}_{\left(p\right)}-x_{p}\right\|}_{\infty }^{\left(n+1\right)}=\underset{j=N_{0},\cdots ,N;j\neq n}{\min}{\left\|x_{p}-x_{j}\right\|}_{\infty }=\underset{j=N_{0},\cdots ,N;j\neq n}{\min}\underset{\;0\leq l\leq n-1}{\max}\left|x_{p+l}-x_{j+l}\right|$. If ${\left\|x_{\eta }{}_{\left(p\right)}-x_{p}\right\|}_{\infty }^{\left(n\right)}\ll {\left\|x_{\eta }{}_{\left(p\right)}-x_{p}\right\|}_{\infty }^{\left(n+1\right)}$, namely, two non-adjacent points become two adjacent points when projected on the low trajectory, such adjacent points are the false neighbors.

Specifically, *x**_η(p)_* is considered as the false nearest neighbor of *x_p_* if it satisfies:(15)$$\left|\frac{{\left\|x_{\eta }{}_{\left(p\right)}-x_{p}\right\|}_{\infty }^{\left(n+1\right)}-{\left\|x_{\eta }{}_{\left(p\right)}-x_{p}\right\|}_{\infty }^{\left(n\right)}}{{\left\|x_{\eta }{}_{\left(p\right)}-x_{p}\right\|}_{\infty }^{\left(n+1\right)}}\right|\geq R_{T}$$
where, *R_T_* is a threshold value, $R_{T}\in \left[10,50\right]$.

With the increase of dimension *n*, the ratio of false neighbors *δ* decreases, where *δ* is the ratio of false neighbors to total neighbors. When *δ* is less than a certain threshold *δ*_0_ or *δ* no longer changes with the change of *n*, then the geometric structure of the state space has been completely opened, and the dimension *n* at this time is the embedding dimension.

Step 2. Calculating the minimum embedding dimension

Firstly, define the variable *α*(*p*,*n*):(16)$$\alpha \left(p,n\right)=\frac{{\left\|x_{\eta }{}_{\left(p\right)}-x_{p}\right\|}_{\infty }^{\left(n+1\right)}}{{\left\|x_{\eta }{}_{\left(p\right)}-x_{p}\right\|}_{\infty }^{\left(n\right)}}$$

Then, define the mean value of *α*(*p*,*n*) with respect to *n* as:(17)$$E\left(n\right)=\frac{1}{N-N_{0}+1}\sum_{n=N_{0}}^{N}\alpha \left(p,n\right)$$

To capture the change of embedding dimension from *n* to *n* + 1, redefine:(18)$$E_{0}\left(n\right)=\frac{E\left(n+1\right)}{E\left(n\right)}$$

The minimum embedding dimension of reconstructed state space can be judged according to the change of *E*_0_(*n*). If it is found that when the embedding dimension increases to a certain extent, *E*_0_(*n*) will no longer change or change to a very small extent, then *n* + 1 is the required minimum embedding dimension. At this time, the corresponding *n_y_* and *n_u_* are the orders of the corresponding model [44]. After using the false nearest neighbor method, the optimal order of the input vector (*n_u_*) and output vector (*n_y_*) of the PWA model are 1 and 2 respectively.

### 2.4. Collaborative Optimization of the Number and Parameters of the Models

In order to improve the prediction performance of PWA multi-models, the number of sub-models and the parameters of each sub-model must be optimized. The PSO is widely adopted for fuzzy system control, neural network training, and function optimization, because it is simple to define and easy to implement, with fast computing speed [45]. Therefore, this paper applies the PSO to determine the number of models and to optimize the parameters of local models in the PWA multi-model, and introduces the ICF to improve the ability of the standard PSO to avoid the local optimum early and perform accurate local search thereafter.

#### 2.4.1. PSO Algorithm

In the standard PSO algorithm, each particle in the swarm is of zero volume in the D-dimensional search space, and is represented by its current position. Each particle flies at a certain velocity in the search space, and keeps adjusting the velocity based on its experience and that of every other particle. 

The current velocity and position of the *i*-th particle can be defined as:(19)$$V_{i}=\left(v_{i,1},v_{i,2},\cdots ,v_{i,D}\right)\;\mathrm{and}\;P_{i}=\left(p_{i,1},p_{i,2},\cdots ,p_{i,D}\right)$$

Then, the individual best-known position of the *i*-th particle and the global best-known position of the swarm can be respectively expressed as: (20)$$P_{i}^{b}=\left(P_{i,1}^{b},P_{i,2}^{b},\cdots ,P_{i,D}^{b}\right)\;\mathrm{and}\;P_{g}^{b}=\left(P_{g,1}^{b},P_{g,2}^{b},\cdots ,P_{g,D}^{b}\right)$$

For the generation *t* + 1, the velocity and position of the *i*-th particle in the *j*-th dimension (1 ≤ *j* ≤ *D*) can be updated by:(21)$$v_{i,j}\left(t+1\right)=wv_{i,j}\left(t\right)+c_{1}r_{1,i}\left(p_{{}_{i,j}}^{b}\left(t\right)-p_{i,j}\left(t\right)\right)+c_{2}r_{2,i}\left(p_{{}_{g,j}}^{b}\left(t\right)-p_{i,j}\left(t\right)\right)$$
(22)$$p_{i,j}\left(t+1\right)=p_{i,j}\left(t\right)+v_{i,j}\left(t+1\right)$$
where, *w* is inertia weight; *c*_1_ and *c*_2_ are acceleration coefficients, both of which are constants; *r*_1,*i*_ and *r*_2,*i*_ are two random functions that change in the range of [0, 1].

In the position update Equation (21) of the standard PSO, if $p_{i,j}=p_{i,j}^{b}=p_{g,j}^{b}$, which means that the current position of particle *i* is the best position it has experienced and also the best position the whole particle population has experienced so far, then the change of $v_{i,j}\left(t+1\right)$ only depends on $wv_{i,j}\left(t\right)$. This implies that when the swarm approaches a local optimal solution, its velocity will approach zero. This means that all particles tend to stop moving near the local optimal solution. Thus, the standard PSO would converge prematurely, and get stuck in the local optimal position. However, there are many local optimal solutions for non-convex problems, and the standard PSO will converge to a local optimum very easily. For these reasons, PSO must be improved.

#### 2.4.2. ICF-PSO Algorithm

To overcome premature convergence, this paper improves the PSO into the ICF-PSO. The position update formula was not changed, while the velocity update formula was refined by introducing the ICF *μ*:(23)$$v_{i,j}\left(t+1\right)=\mu \left(wv_{i,j}\left(t\right)+c_{1}r_{1,i}\left(p_{{}_{i,j}}^{b}\left(t\right)-p_{i,j}\left(t\right)\right)+c_{2}r_{2,i}\left(p_{{}_{g,j}}^{b}\left(t\right)-p_{i,j}\left(t\right)\right)\right)$$
where, $\mu =\delta^{\sigma^{{}^{\frac{G}{N+G}}}-1}$ is the compression factor, which both restrains the velocity of particles and enhances the local search ability of the algorithm. Note that *δ* and *σ* are constants in *μ*, both of which are key to the control of the constraint effect of the compression factor; *N* is the total number of iterations; *G* is the current number of iterations. The rationale for the improved compression factor is as follows. In this paper, the characteristic curve of *μ* is made to be a monotonically decreasing curve in the first quadrant as the number of current iterations G increases by determining the appropriate values of *δ*, *σ* and *N*. This curve must contain the following characteristics, that is, the value of *μ* must vary between 1 and 0; the value of *μ* must decrease rapidly at the beginning of the iteration, and then the rate of decrease of the value of *μ* tends to smooth out as the iteration proceeds to the middle and later stages. The compression factor satisfying the above conditions is substituted into the velocity update formula of the standard PSO algorithm, which makes the particle velocity decrease continuously during the iteration process under the constraint of the compression factor. The particle velocity satisfies the following characteristics during the decreasing process, the particle can maintain a fast velocity at the beginning of the iteration, which can make the particle jump out of the local optimum effectively with the fastest speed, but the rate of particle velocity decreasing is fast during this time. As the iteration proceeds to the middle and later stages, the particle velocity becomes small and the particle velocity decreases very little and smoothly at this time. This kind of particle is able to search the global optimum precisely and smoothly, just like the carpet search of radar. The above is the rationale for the improved compression factor *μ*. This paper will describe in detail how to determine reasonable values of *δ*, *σ* and *N*, in the following subsections.

#### 2.4.3. Determination of Parameter Values in ICF

This paper tries to maximize the efficiency and optimization effect of the compression factor by determining the value range of *δ* and *σ*. Firstly, the total number of iterations *N* was assumed to be 300. Secondly, both *δ* and *σ* must be positive numbers, according to mathematical knowledge (otherwise, *μ* will be meaningless). Finally, the value of the compression factor should decrease with the increase of *G*, judging by its functional features. Therefore, the value range of *δ* and *σ* could only be one of the two cases: (1) δ∈(0,1) and σ∈(1,+∞); (2) δ∈(1,+∞) and σ∈(0,1). The variation curves of *μ* in these two cases are discussed in detail below:

(1)δ∈(0,1) and σ∈(1,+∞)


Figure 2a shows the curve of *μ* with the increase of *σ* at *δ* = 0.5, and Figure 2b shows the curve of *μ* with the increase of *δ* at *σ* = π. It can be seen that when the value of *δ* was constant, the constraint performance of ICF *μ* was better at a larger *σ* value; when the value of *σ* was fixed, the smaller the value of *δ* was, the better the constraint performance was. The values of *σ* and *δ* should be considered comprehensively to optimize the constraint performance of the ICF *μ*. Experimental results show that, in case (1), the constraint performance of *μ* is the best at *δ* = 0.1 and *σ* = 2π (Figure 2c).

(2)δ∈(1,+∞) and σ∈(0,1)

Figure 2d shows the curve of *μ* with the increase of *σ* at *δ* = 2π, and Figure 2e shows the curve of *μ* with the increase of *δ* at *σ* = 0.5. It can be seen that when the value of *δ* was unchanged, the constraint performance of ICF *μ* was better at a smaller *σ* value; when the value of *σ* was fixed, the larger the value of *δ* was, the better the constraint performance was. The values of *σ* and *δ* should be considered comprehensively to optimize the constraint performance of the ICF *μ*. Experimental results show that, in case (2), the constraint performance of *μ* is the best at *δ* = 24 and *σ* = 0.01 (Figure 2f).

To sum up, the compression factor *μ* has an optimal state in either case, depending on the values of *δ* and *σ*. Here, the *δ* and *σ* values in compression factor *μ* are set to 24 and 0.01, respectively.

In fact, the concept of compression factor *φ* was put forward by Clerc et al. in as early as 1999 [46]:(24)$$\varphi =\frac{2}{\left|2-\rho -\sqrt{\rho^{2}-4\rho }\right|}\;\;\mathrm{and}\;\rho =c_{1}+c_{2}$$
where, *ρ* > 4 and *φ* is the compression factor; *c*_1_ and *c*_2_ are acceleration constants. The problem is that the compression factor is too rigid [47]. The value of *φ* only depends on the size of *c*_1_ and *c*_2_, failing to consider the iterative effect. With the elapse of time, such a compression factor is no longer suitable for optimizing complex nonlinear systems. Compared with the original compression factor *φ*, the improved compression factor *μ* proposed by the authors can change iteratively, and adapt its constraint effect to the specific optimization task, in the light of the combined effect of factors like *δ*, *σ*, *N* and *G*.

The details of the performance analysis of the ICF-PSO algorithm are shown in the Appendix A.

#### 2.4.4. ICF-PSO Algorithm Applied to Optimize the Number and Parameters of PWA Local Models

The multi-objective optimization Problem (9) cannot be easily solved by the traditional phased strategy, because of the strong coupling between the two objective functions. Drawing on the idea of non-inferior optimal solution, this paper chooses to optimize objective function *J*_1_ by the step-by-step iterative method, in view of the fact that *J*_1_ depends heavily on *J*_2_ but *J*_2_ is independent of *J*_1_: First, select the non-optimal solution *S* that satisfies *J*_2_ and determine an optimal search threshold *Y_ts_* for *J*_1_. Then, *S* is substituted into *J*_1_, and *J*_1_ is optimized to obtain the optimal solution of the corresponding local model parameter; if the optimized *J*_1_ cannot meet the preset threshold *Y_ts_*, optimize *S* and re-optimize *J*_1_. Repeat these steps until the threshold *Y_ts_* is satisfied.

To optimize objective function *J*_2_, it is critical to reasonably divide the input space *Ω_N_* into *S* subspaces that satisfy conditional Equation (25), so as to determine the number of local models:(25)$$\sum_{i=1}^{S}M_{i}=N$$
where, *M_i_* is the number of input vectors contained in the *i*-th subspace; $\Omega_{i}\left(i=1,2,\cdots ,S\right)$ is the *i*-th input subspace.

After the input space dividing *Ω**_i_* is obtained, the corresponding sample dataset can be determined by:(26)$$Z_{i}=\left\{z\left(k\right)|X\left(k\right)\in \Omega_{i},\;y\left(k\right)=f\left(X\left(k\right)\right)\right\}$$
where, $\overset{S}{\underset{i=1}{\cup }}X_{i}=Z_{N}$; *Z_i_* is the sample dataset corresponding to *Ω_i_*, which uniquely determines the modeling space of *f* (*X*(*k*)).

When the system model is unknown, *y*(*k*)∈*R* can be replaced by the actually observed output. The difficulty of *J*_2_ is that the division of input space cannot be separated from the parameter identification of the local model. The strong coupling between the two objective functions makes it necessary to determine the number of models and estimate the model parameters simultaneously. Therefore, this paper employs the ICF-PSO algorithm, which excels in parallel search and data clustering to optimize the number of models and parameters.

During the optimization, the modeling parameters reflecting each local model are combined with the corresponding cluster center into a particle:(27)$$P=\left\{\theta_{1},Z_{1},\cdots ,\theta_{i},Z_{i},\cdots ,\theta_{S},Z_{S}\right\}$$
where, $Z_{i}\left(i=1,2,\cdots ,S\right)$ is the cluster center of the *i*-th cluster.

By the ICF-PSO algorithm, the study optimized the number of local models and the parameters of each local model at the same time [48]. The detailed steps are as follows:(1)Step 1. Initialization

Set the number of non-optimal models *S* = 1, swarm size *N_S_*, inertia weight *w*, acceleration coefficients *c*_1_ and *c*_2_, the threshold *Y_ts_* of objective function *J*_1_, maximum allowable number of iterations *N_it_*, initial position *P_i_* and initial velocity of each particle *V_i_*, etc.

(2)Step 2. Data clustering

Because the PWA system is locally linear, the data close to each other are very likely to fall into the same class. Thus, each local subspace *Z_i_* can be divided by the distance between the data and the given cluster center. The specific method is to calculate the norm distance from each modeling space vector *z*(*k*) to a given cluster center $\overset{\wedge }{Z_{i}}$; if the modeling space vector is the closest to a given cluster center, it will be assigned to that cluster:(28)$$z\left(k\right)\in Z_{i}\;\mathrm{and}\;I=\arg\underset{k}{\min}{\left\|z\left(k\right)-\overset{\wedge }{Z_{i}}\right\|}_{2}$$
where, i=1,2,⋯,S.

(3)Step 3. Initial fitness evaluation

Evaluate the initial fitness of each particle according to the objective function *J*_1_.

(4)Step 4. Evolution of particle swarm

Update the position and velocity of particles according to Equations (22) and (23), respectively.

(5)Step 5. Fitness update

Cluster the data by Step 2, and update the fitness of each particle by objective function *J*_1_.

(6)Step 6. Termination conditions(1)If the maximum number of iterations is not reached and the threshold of the objective function is not met, return to Step 3.(2)If the maximum number of iterations is not reached, but the threshold of the objective function is met, terminate the algorithm.(3)If the maximum number of iterations is reached and the threshold of the objective function is met, terminate the algorithm.(4)If the maximum number of iterations is reached, but the threshold of the objective function is not met, go to Step 7.
(7)Step 7. Make *J*_1_ = *S* + 1, return to Step 1, and restart the search

The above steps can be organized into the following flow chart in Figure 3.

### 2.5. Determination of Dividing Surfaces

So far, the sample space has been divided into *S* local spaces *Ω_i_* (*i* = 1, 2, ···, *S*). According to the definition of the PWA system in Equation (2), the convex polyhedrons with different scopes have no overlapping parts except the common boundary. Therefore, the problem of finding the optimal dividing surface is directly transformed into the linear division of cluster data. The least square support vector machine (LS-SVM) algorithm was selected to solve the optimal dividing surface between local PWA models. The selection was made because the LS-SVM is a classical classifier to handle small samples, and high-dimensional nonlinear data. The training speed of the model is accelerated, owing to the combination between the standard SVM and the least squares method.

#### 2.5.1. Least Square Support Vector Machine

The core idea of LS-SVM is that in a given sample data set ${\left\{x_{k},y_{k}\right\}}_{k=1}^{N}$, in which *x_k_*∈*R^n^* and *y_k_*∈{−1, 1}. To find the optimal classification hyperplane $w^{T}x+b=0$, that is, to solve the following optimization problems [49], namely
(29)$$\begin{array}{l} \min J\left(w,e\right)=\underset{w,b,e}{\min}\frac{1}{2}w^{T}w+\mu \sum_{k=1}^{N}e_{k}^{2}\\ s.t.y_{k}\left[w^{T}\varphi \left(x_{k}\right)+b\right]=1-e_{k}\;\;\;k=1,2,\cdots ,N \end{array}$$
where, $\varphi (g):R^{n}\to R^{n_{H}}$ is a function that maps input data to a high-dimensional feature space; weight vector $w\in R^{n_{H}}$; error variables and offset values satisfy $e_{k}\in R$ and b∈R; μ>0 is the weight coefficient, which can make LS-SVM find the optimal hyperplane and ensure the minimum deviation [50].

Introducing Lagrange function into Equation (29), then
(30)$$L\left(w,b,e;\alpha \right)=J\left(w,e\right)-\sum_{k=1}^{N}\alpha_{k}\left\{\left[y_{k}\left(w^{T}\varphi \left(x_{k}\right)+b\right)-1+e_{k}\right]\right\}$$
where, $\alpha_{k}\geq 0$ is Lagrange multiplier; *x_k_* is the support vector.

By deriving *w*, *b*, *e_k_*, *α_k_* in the above formula respectively, the following linear equation system about *α* and *b* can be obtained:(31)$$\left[\begin{array}{cc} 0 & y^{T}\\ y & \Omega +I/\mu \end{array}\right]\left[\begin{array}{l} b\\ \alpha \end{array}\right]=\left[\begin{array}{l} 0\\ 1_{v} \end{array}\right]$$
where, *y* = [*y*_1_,⋯,*y_N_*]; *α =* [*α*_1_,⋯, *α_N_*]; 1*_v_* = [1,⋯,1]; *Ω* is called a kernel matrix, and its expression is
(32)$$\Omega_{i,j}=y_{i}y_{j}\varphi {\left(x_{i}\right)}^{T}\varphi \left(x_{j}\right)=y_{i}y_{j}K\left(x_{i},x_{j}\right)$$
where, *i,j* = 1,⋯,*N*. A set of *α* and *b* can be obtained by solving Equation (31). Finally, the classification hyperplane obtained by LS-SVM is shown below:(33)$$f\left(x\right)=sgn\left[\sum_{k=1}^{N}\alpha_{K}y_{k}K\left(x,x_{k}\right)+b\right]$$
where, the kernel function must satisfy Mercer condition. The kernel function of this paper uses RBF kernel function. The expression of the RBF kernel function is shown below.
(34)$$K\left(x_{i},x_{j}\right)=\exp\left(-\frac{{\left\|x_{i}-x_{j}\right\|}^{2}}{\sigma^{2}}\right)$$
where, *σ* is the width parameter of the function, which controls the radial range of action of the function.

#### 2.5.2. LS-SVM Applied to Determine the Parameters of Dividing Surfaces

To begin with, a sample space is defined:(35)$$\;Z_{i}={\left\{z\left(k\right)|X\left(k\right)\in \Omega_{\;i},\;y\left(k\right)=f\left(x\left(k\right)\right)\right\}}_{k=1}^{M_{i}}$$
where, $Z_{N}=\overset{S}{\underset{i=1}{\cup }}Z_{i}$.

The LS-SVM determines the dividing surface in two steps: First, find out the adjacent subspaces; Then, determine the parameters of dividing surfaces between adjacent subspaces. The specific steps are as follows:

Step 1. Determine the modeling subspaces adjacent to each other by the nearest neighbor rule of cluster center, namely:(36)$$\left\{Z_{i},Z_{j}\right\}=\underset{2\leq i,j\leq s,i\neq j}{\min}\left\{{\left\|Z_{i}-Z_{j}\right\|}_{2}\right\}$$

Step 2. Determine the parameters of the dividing surface between two adjacent subspaces.

According to Equation (4), the dividing surface of two adjacent subspaces can be described as: $h_{i,j}=\left\{w_{i,j}^{T}X+v_{i,j}=0\right\}$. Suppose *y*(*i*) = 1 for the sample belonging to *Ω**_i_* subspace, and *y*(*j*) = −1 for the samples belonging to *Ω_j_* subspace. Then, kernel matrix *Ω* in Equation (31) can be transformed by LS-SVM into:(37)$$\Omega_{i,j}=y_{i}y_{j}\varphi {\left(X_{i}\right)}^{T}\varphi \left(X_{j}\right)=y_{i}y_{j}K\left(X_{i},X_{j}\right)$$

Without changing other parameters in Equation (31), *N* groups of *α* and *b* can be obtained by solving the new linear equation set. Finally, the dividing surface can be obtained by LS-SVM as:(38)$$f\left(x\right)=sgn\left[\sum_{i=1}^{N}\alpha_{i}y_{i}K\left(X,X_{i}\right)+b\right]$$
where, *N* = *M_i_* + *M_j_* is the total number of vectors in subspace *Ω**_i_* and subspace *Ω_j_*.

Accordingly, the parameters of dividing surfaces can be obtained by:(39)$$w_{i,j}=\sum_{i=1}^{N}\alpha_{i}y_{i}X_{i}\;\;\mathrm{and}\;v_{i,j}=y_{i}-w_{i,j}^{T}X_{i}$$

Therefore, the parameters of dividing surfaces can be expressed as $h_{i,j}=\left[w_{i,j}^{T},v_{i,j}\right]$.

### 2.6. Modeling Steps of the PWA Soft Sensor Model

Let *D* be the sample dataset of the fermentation process. The sample dataset is a kind of MISO Problem (1). Then, the model identification problem is transformed into an MIQP optimization Problem (8). The optimal solution of the MIQP problem can be obtained by the following steps:(1)Step 1. Determining model order

The FNN is introduced to determine the embedding dimension in the state space reconstruction of nonlinear time series in the fermentation process. The false nearest neighbor can be identified by Equation (15). When the dimension *n* increases continuously, the ratio *δ* of the false nearest neighbor will approach a threshold. The dimension at this very moment is taken as the embedding dimension. The change of embedding dimension is evaluated by *E*_0_(*n*) in Equation (18). When the embedding dimension *n* = *n_y_* + *n_u_* increases to a certain extent, *E*_0_(*n*) tends to be stable. At this moment, the *n_y_* and *n_u_* are the order of the corresponding model.

(2)Step 2. Collaborative optimization of the number and parameters of models

Because of their strong coupling, *J*_1_ and *J*_2_ must be determined synchronously. Therefore, the powerful parallel search ability of ICF-PSO algorithm is utilized to optimize the multi-objective optimization Problem (9). By the ICF-PSO, Equations (22) and (23) are updated, such that the particles can avoid the local optimum.

(3)Step 3. Determining dividing surfaces

The adjacent modeling subspaces {*Z_i_*, *Z_j_*} are determined by the nearest neighbor rule of cluster center, and the dividing surfaces of adjacent subspaces are determined by $h_{i,j}=\left\{w_{i,j}^{T}X+v_{i,j}=0\right\}$. By solving the new linear equation set, *N* groups of *α* and *b* are obtained. Then, the parameters $w_{i,j}^{T}$ and $v_{i,j}$ of dividing surfaces can be solved by Equation (38), namely, $h_{i,j}=\left[w_{i,j}^{T},v_{i,j}\right]$.

The modeling process of the PWA algorithm is illustrated in Figure 4.

### 2.7. Introduction of Experimental Work

The online control of *Pichia pastoris* fermentation is a difficult problem, due to the complexity of the fermentation process, and the cost of online detection instruments for key biomass variables being very high. To provide information for the online control and to optimize the fermentation process, it is significant to establish real-time soft sensor models for cell concentration *X* and protease K concentration *P*.

The sample data used in this paper all came from the microbial fermentation lab of Zhenjiang Yangzhong Weikete Bioengineering Equipment Co., Ltd. (Zhenjiang, China), and the fermentation experiment equipment adopted RTY-C−100 L model. Taking *Pichia pastoris* fermentation as the object, the *Pichia pastoris* KM71, Mut^S^ constructed by the Key Laboratory of Animal Husbandry and Veterinary Institute of Shanghai Academy of Agricultural Sciences, was selected as the strain, and the expression vector and foreign genes were pPICZαA and IFNαcDNA [51]. The experiment was carried out according to the schematic diagram of the *Pichia pastoris* fermentation process (Figure 5). After deeply analyzing the fermentation process, the temperature of fermentation broth *T*, dissolved oxygen concentration *DO*, pH of fermentation broth, air flow *q*, stirring speed *v*, and the pressure of fermentation tank *p* were selected as auxiliary variables. In order to clarify the auxiliary and key biomass variables of the ICF-PSO-PWA model at a glance, a model expression with a generalized form for the input and output variables is given in Equation (39).
(40)φ(X,P)=f(T,Do,pH,q,v,p)
where, f(g) is a generalized form of the complex nonlinear relationship between the auxiliary variables in the ICF-PSO-PWA model; φ(X,P) is any one of *X* and *P*. That is, the auxiliary variables (*T*, *Do*, pH, *q*, *v*, *p*) are the input variables of the model we use, and the key biomass variables (*X*, *P*) are the output variables we need.

The initial values of the input variables in the fermentation process were set as follows: the pressure of the fermentation tank was controlled at 0.04 Mpa, the stirring speed of the motor was about 250 rpm, the fermentation temperature was controlled at about 28 ± 0.5 °C, and the air flow was controlled at the range of 1000–1200 L/h. The oxygen content was maintained between 35% and 45%, and the pH was 7.3.

Under normal fermentation conditions, the environmental variables of the fermentation processes were monitored by sensors every 5 min and uploaded to the host computer. The cell concentrations and protease K concentrations were measured offline every 1 h. The cell concentration was measured by the American Lehman Lamotte 1200 photoelectric colorimeter, and the concentration of proteinase k was measured by the UV−260 model automatic spectrophotometer of Shimadzu Corporation, Japan. The adjustment, logarithmic growth and stabilization phases in the fermentation processes were selected as the data collection time period, which was approximately the first 80 h of the fermentation process. Finally, 10 batches of sample data were extracted to apply our algorithm. Subsequently, 6 out of these 10 batches were used for training, the 7th and 8th batch for online correction of the initial PWA model, and the 9th and 10th batch for verifying the effectiveness and prediction accuracy of the PWA model. As the output variables are measured offline once every 1 h and the input variables once every 5 min, there is no one-to-one correspondence between the two, so the interpolation methods are used to convert the output variables offline every 1 h to the input variables every 5 min, forming a one-to-one correspondence. The data are then pre-processed by a coordinate transformation, which normalizes all sample data to between −1 and 1, making the calculation easier and faster. Through coordinate transformation and interpolation operation on 10 batches of sample data, a total of 960 sets of sample data were obtained at last, including 576 sets of sample data in training set, 204 sets of sample data in verification set and 180 sets of sample data in test set.

## 3. Results and Discussion

After determining that the order of input vector (*n_u_*) is 1 and the order of output vector (*n_y_*) is 2, *X^T^*(*k*) = [*u^T^*(*k*), *y^T^*(*k*), *y^T^*(*k* + 1)] is selected to form the input vector and the local model is chosen to be of the form *y*(*k*) = [*y^T^*(*k* − 1), *u^T^*(*k* − 1), 1]*θ*. The dividing surface equation is *h_i,j_*[*y^T^*(*k* − 1), *u^T^*(*k* − 1), 1]*^T^* = 0, which separates the *i*-th local model from the region where the *j*-th local model is located. After using the ICF-PSO algorithm to optimize the PWA model, the number of local models is *S* = 3, which exactly corresponds to the three phases of the sample data in this paper: adjustment, logarithmic growth and stable phase. The local model parameters *θ_i_* (*i* = 1, 2, 3) and the parameters of dividing surfaces *h_i,j_* are shown in the Table 1.

To verify its performance, the ICF-PSO-PWA model was compared with the PSO-WLSSVR model based on global modeling, which used Gaussian radial basis kernel (RBF) function $k(x_{1},x_{2})=\exp(-{\left\|x_{1}-x_{x}\right\|}^{2}/\sigma^{2})$, with *σ* = 0.2, and penalty coefficient *C* = 10. The WLS-SVR model introduces the least squares algorithm and weights on the basis of the SVR model, which greatly enhances the prediction accuracy and generalization ability of the SVR model. It is a very good comparator. All simulation results were obtained based on MATLAB R2019b software.

Figure 6a,b present the prediction results of soft sensor models based on ICF-PSO-PWA and PSO-WLSSVR, respectively. Obviously, the fitting effect of ICF-PSO-PWA soft sensor model is better than that of the PSO-WLSSVR model, regardless of the bacterial concentration or the concentration of protease K. In particular, the fitting effect of the PSO-WLSSVR model is clearly inferior to the ICF-PSO-PWA model in terms of the cell concentration in the later stage.

Figure 7a,b compare the relative error of cell concentration and protease K concentration based on ICF-PSO-PWA and PSO-WLSSVR, respectively. For both ICF-PSO-PWA and PSO-WLSSVR soft sensor models, the prediction errors gradually decrease with time and are eventually stabilized. However, the ICF-PSO-PWA soft sensor model has a much smaller error range, a relatively high accuracy in the whole sample interval, and an obviously smaller final error than the PSO-WLSSVR model.

Table 2 shows the mean absolute percentage error (MAPE) of cell concentration and proteinase K concentration based on ICF-PSO-PWA and PSO-WLSSVR. It can be seen from Table 2 that compared with the PSO-WLSSVR soft sensor model, the ICF-PSO-PWA soft sensor model has an improved accuracy by 2.4642% when predicting the cell concentration, and an improved accuracy by 6.5127% when predicting the proteinase K concentration. The mean prediction accuracy of ICF-PSO-PWA soft sensor model has been improved by 4.4884%.

Furthermore, the root mean square errors (RMSEs) of the ICF-PSO-PWA and PSO-WLSSVR soft sensor models were tested on different scale datasets. The test results in Table 3 show that, with the growing number of test samples, the RMSE of the PSO-WLSSVR soft sensor model increases, while that of the ICF-PSO-PWA model does not change significantly. This is because the latter model divides the working interval of the system into several subintervals, and models each subinterval separately; besides, multiple local models are used in place of the global single model, making full use of the information of each sample. As time goes by, the ICF-PSO-PWA model can adapt to the changing working conditions, and achieve excellent generalization performance and self-adaptability.

## 4. Conclusions

In order to solve the problem that the key biological variables in the fermentation process of *Pichia pastoris* are difficult to directly measure online, a PWA multi-model soft sensor modeling method based on the idea of “decomposition-synthesis” was proposed to measure the key variables in the fermentation process of *Pichia pastoris* online. In order to identify the PWA multi-model, the FNN was used to solve the order of the input and output vectors of the model. Then, a novel ICF-PSO algorithm was proposed for collaborative optimization of the number of local models and the parameters of each local model. Finally, the LS-SVM was used for the determination of the parameters of the dividing surfaces in the modeling process, so as to delineate the regions where each local model plays a role. MATLAB simulation results show that the prediction accuracy of the ICF-PSO-PWA model for X and P reach 1.8932% and 2.5974%, respectively, compared with 4.3573% and 9.1101% of the prediction accuracy of the PSO-WLSSVR model for X and P, the prediction accuracy of the ICF-PSO-PWA model improved by 2.4642% and 6.5127%. The RMSEs of the ICF-PSO-PWA model for X and P prediction results remain at 0.015402352 and 0.711287476, respectively, which are smaller than 0.033457354 and 2.278762339 of the PSO-WLSSVR model, indicating that the prediction results of the ICF-PSO-PWA model are more stable. The simulation results above demonstrate the effectiveness of the ICF-PSO-PWA soft sensor method in predicting the key biological variables in the fermentation process of *Pichia pastoris*. The proposed model provides a feasible theoretical approach to solve the soft sensing of key biological variables in the *Pichia pastoris* fermentation process.

The PWA modeling method, as a special form of modeling method, partitions the nonlinear complex system into several linear systems, so that the system can be solved using the existing mature linear theory, breaking through the bottleneck that the traditional single model is extremely complex and cannot meet the needs of actual engineering when the global description of increasingly complex controlled objects is performed. The strong nonlinear approximation capability and simple model structure of PWA are not only loved by scholars in the field of modeling, but also a hot research topic in the control community. In the future, we are interested in studying the optimal control of PWA models and the design of online monitoring systems for the whole fermentation process.

## Figures and Tables

**Figure 1 sensors-21-07635-f001:**
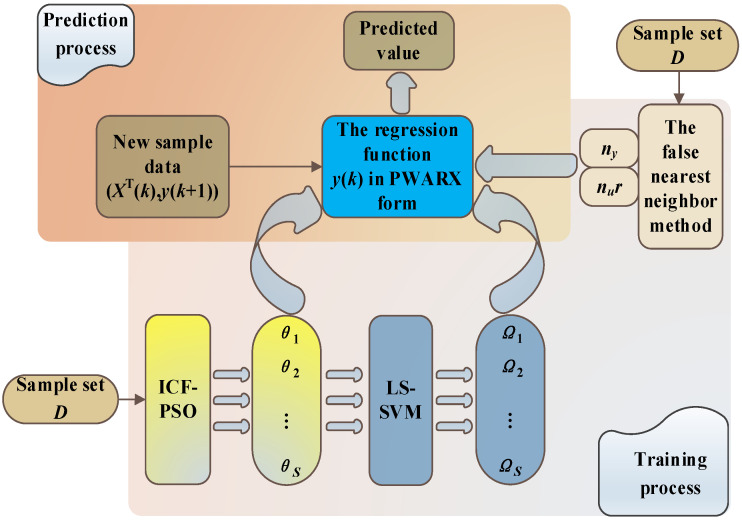
Overall structure diagram of modeling algorithm.

**Figure 2 sensors-21-07635-f002:**
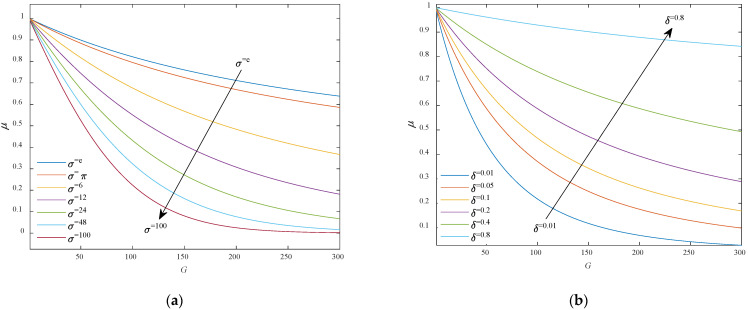
The characteristic curve of *μ* when the values of *δ* and *σ* change, including (**a**) *δ* = 0.5 and *σ* was increasing between (1,+∞), (**b**) *σ* = π and *δ* was increasing between (0.1), (**c**) one of the best constraint performance of *μ* was at *δ* = 0.1 and *σ* = 2π, (**d**) *δ* = 2π and *σ* was increasing between, (**e**) σ = 0.5 and *μ* was increasing between (1,+∞), (**f**) one of the best constraint performance of *μ* was at *δ* = 24 and σ = 0.01.

**Figure 3 sensors-21-07635-f003:**
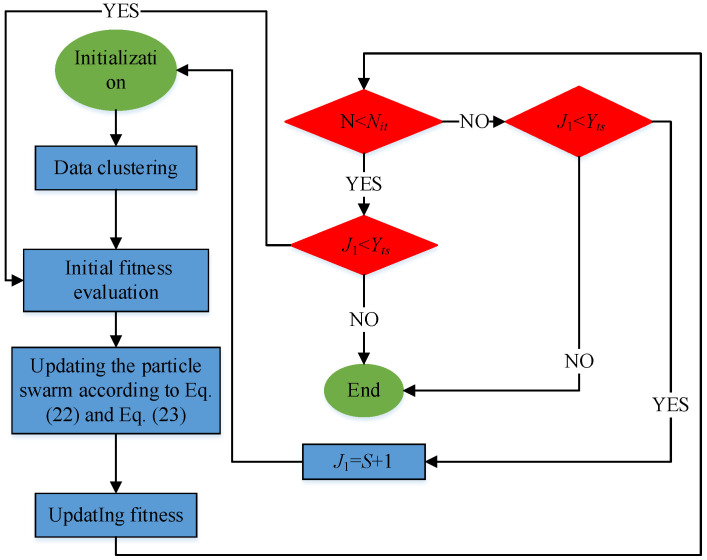
Flow chart of ICF-PSO for optimizing the number and parameters of PWA local models.

**Figure 4 sensors-21-07635-f004:**
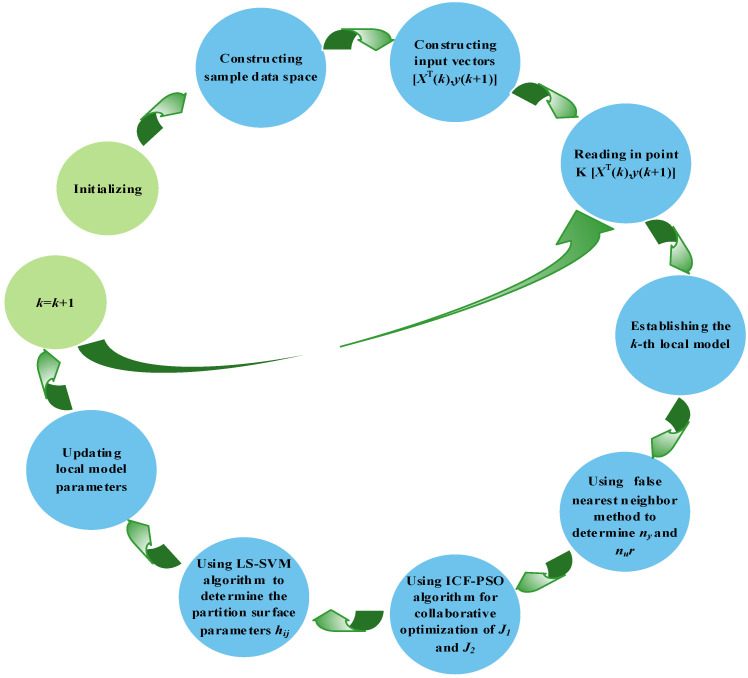
Flow chart of ICF-PSO-PWA soft sensor modeling algorithm.

**Figure 5 sensors-21-07635-f005:**
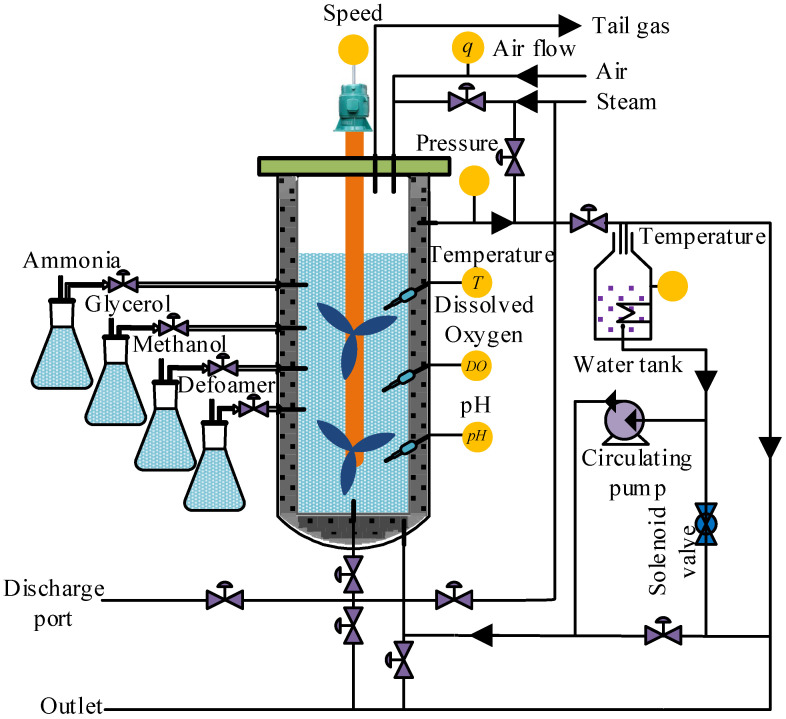
Schematic diagram of *Pichia pastoris* fermentation process.

**Figure 6 sensors-21-07635-f006:**
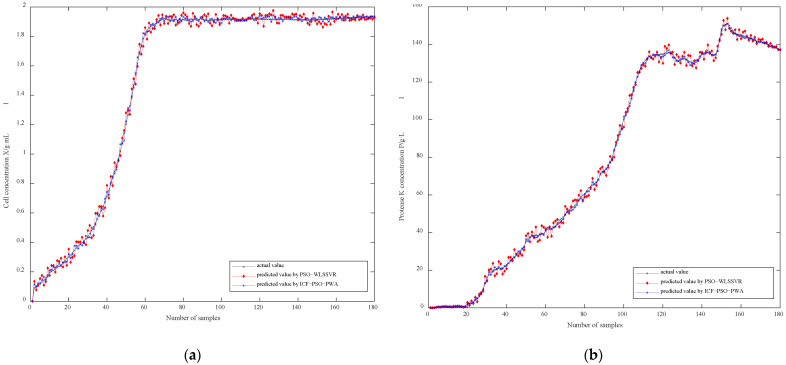
Comparison curve between the soft sensor value and the actual value of (**a**) cell concentration and (**b**) protease K concentration.

**Figure 7 sensors-21-07635-f007:**
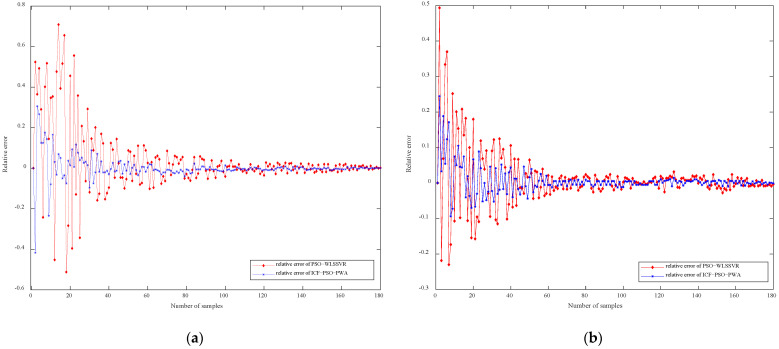
Comparison curve of relative error of (**a**) cell concentration and (**b**) protease K concentration.

**Table 1 sensors-21-07635-t001:** Local model modeling parameters.

	The Local Model Parameters	The Parameters of Dividing Surfaces
Local model I	*θ*_1_** = [0.5957, −0.1847, −0.0964, 0.4447, 0.3635, 0.0464, 0.5567, −0.2888] *^T^*	*h*_13_** = [1, 1.0445, −0.0045, 0.9854, 0.5421, −0.2347, −1.0786, 0.0864]
Local model II	*θ*_2_** = [0.0962, 0.9649, 0.0434, 0.0762, 0.9564, 0.0615, −0.6332, 1] *^T^*	
Local model III	*θ*_3_** = [0.1629, 0.9885, −0.0003, 1, 0.1005, −0.6575, 0.3209, 0.6887] *^T^*	*h*_23_** = [1, −1.8753, 0.9753, −0.8656, 1.9531, 0.6542, −0.5423, 1.0112]

**Table 2 sensors-21-07635-t002:** Comparison of prediction accuracy of soft sensor models.

	PSO-WLSSVR	ICF-PSO-PWA	Accuracy Improvement
MAPE of cell concentration	0.043573488	0.018931913	2.4642%
MAPE of protease K concentration	0.091100816	0.025974078	6.5127%
Mean accuracy improvement	4.4884%		

**Table 3 sensors-21-07635-t003:** RMSE of different test sample sets.

Number	PSO-WLSSVR		ICF-PSO-PWA	
	*X*	*P*	*X*	*P*
20	0.034753378	0.242421228	0.013863265	0.057610602
40	0.041227412	1.726108115	0.017189875	0.537857149
60	0.043646458	2.086190315	0.021352118	0.598752504
80	0.041167894	2.272430942	0.019573828	0.635771991
100	0.039357834	2.402638746	0.018463627	0.710257895
120	0.036679143	2.35714544	0.017360173	0.70555966
140	0.035563714	2.38677746	0.016815066	0.745174526
160	0.035008022	2.372930488	0.016138098	0.716623673
180	0.033457354	2.278762339	0.015402352	0.711287476

## Data Availability

Not applicable.

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
