# Peer review of "Study on Multi-Model Soft Sensor Modeling Method and Its Model Optimization for the Fermentation Process of Pichia pastoris"

_sensors, 2021, doi:10.3390/s21227635_

Round 1

Reviewer 1 Report

This paper presents a novel mathematical model for a bioprocess. The authors propose an autoregressive model for the process and divide the entire state space into regions with different set of parameters. In this way the linear ARX could capture the evolution of the bioprocess much better than a simple linear model. Further, the authors also propose an efficient algorithm to estimate the parameters for this model.  

While the authors have made a commendable effort in developing a mathematical model for the system and improving the estimation procedure, there are serious questions/concerns that I have about the article in its present form. I recommend accepting the article only after major revisions are made, particlularly in presentation. 

Comments/Questions

  1. My apologies for being direct about this. But, the efforts made towards writing this manuscript seem much less compared with the effort made towards performing the research. I would strongly recommend going through the article several times clearing up the unnecessary information, organizing the thoughts along the main theme of the article and cleaning typos. Please note, I am not advocating the inclusion of technical jargons, but insisting on only including what is relevant to the story conveyed here. This is the only way you could get your target audience to read this article till the end.
    1. Line 13: PWA. Piecewise affine what? Piecewise affine models? Piecewise affine functions?
    2. Line 31: I am not sure if protein yields are measured in liters.
    3. Line 34, 35: There are two separate phrases in this sentence mentioning 'advantages'. Maybe remove the phrase: 'The advantages above make'.
    4. Line 45: What is meant by 'strongly' non-linear? A function is either linear or non-linear. Strong non-linearity may not be a standard phrase.
    5. Line 58, 59: It is not clear as to what the authors mean by 'soft-sensor technology works in harmony with other computer software'. Does the authors imply that we do not need additional hardware or software to implement soft sensors?
    6. Line 59: Stability - is a term that has a very clear definition in control theory and one cannot make a random statement that all soft sensors are 'stable'. There are several other instances where the authors mention that the optimization algorithm finds global solutions 'stably'. Unless there is a clear reason, please avoid using the word 'stable', particularly in an article proposing a digital twin. 
    7. Please read the whole introduction again and try to organize the thoughts such that it contains (only) what is required. 
    8. Line 191: 'Piecewise inequality' - Again I am not really sure if piecewise inequality is a standard term. The authors mean they are a set of inequalities defining the polyhedron? Why not just call it an 'inequality'?
    9. Line 198: My pdf shows unidentified characters in the function f( , ). Please check if there are any typos. (This could be an issue with my pdf reader as well)  
    10. Line 203, 204: The three steps (in fact four items mentioned here) seem to be the set of problems solved in this work, not the MIQP. Please make corrections if required.   
    11. Eq 11: The upper case 'L' seem to be used in place of dots. This is repeated several times in the pdf. Is this a standard practice? If not, please consider changing this at all places. 
    12. Eq 13: Though 'l' is defined as a decision variable on the right hand side, it does not appear in the function. Please verify. 
    13. Eq 15: Is this equation the definition of R? Please add explanations for what is R, if required. 
    14. Line 247: Is 'm' defined? Did you mean 'n'?
    15. Line 247: 'delta' is already used in the MIQP in a different context. Also, I do not see any definition for delta here. Please verify if there has to be some explanation added. 
    16. Line 495: Please expand RBF if this is not already given in the article before this line. 
    17. Line 585: Please verify if the terms 'key variables' and 'auxiliary variables' are used to refer to input and output variables of the bioprocess. The latter naming convention might be more popular.  
  2. These are just few examples of the potential avenues for improvement. Please go through the entire article and make sure that the notations and explanations are precise.
  3. MIQP - Equation 8. 
    • Line 203: As far as I understand, the MIQP strictly solves only one of these problems - identifying theta. The other problems are hyperparameter identification steps and are probably not MIQPs. Please make changes if required. 
    • In the MIQP, please mention the decision variables beneath the 'min'. That might make this more clear.
    • Why do you need the constraint y(k) = f(X(k))? There is no definition for f (f_hat is defined earlier and that is used in the objective function as well). f seems to be a random function here that is not used elsewhere.
  4. Performance analysis of the ICF-PSO algorithm
    • The primary objective of the paper is to present a soft sensor for the bioprocess. An analysis of the PSO algorithm used here may not be an integral part of this idea and in fact, could deviate the attention of the reader form the main theme. I suggest moving the detailed analysis of the algorithm to appendix.
  5. Global optimality of the solution:
    • Line 285: The 'global best known position' (mentioned in line 283) need not be the 'global optimum'. It may not be possible to quickly move from the argument in line 283 to that in line 285.
    • There are several other instances where the authors mention about globally optimal solutions. If the authors imply that global minimum is always identified, I would have to defer with the authors about this claim over the 7 step algorithm described in Section 2.3.3, due to the following reasons. 
      1. For a non-convex problem, though the PSO does make attempts to evade local optimality (and is successful in many instants), the algorithm in its standard form does not guarantee convergence to global optimum. Even if the PSO could potentially find a globally optimal solution to the parameters,
      2. The update scheme for cluster centers proposed in this paper is accompanied with no explanation for a guaranteed convergence to global optimum.
    • For this reason, I would strongly recommend to abstain from claiming that the algorithm given here would converge to a global optimum unless the authors have a clear reason to support their argument. 
  6. Line 531: Question:  In the 'collaborative optimization' of the number of parameters, is the ICF-PSO algorithm used for solving an MIQP? If yes, is that a convex problem? If so, please explain the rationale behind using a metaheuristic (the PSO) to optimize the problem over a gradient based solver that could solve the problem with guaranteed convergence rates?
  7. Line 641: '... different scale datasets'. These were experimental data or simulated data? If simulated, please explain how these were generated.

Reviewer 2 Report

The .docx file contains my suggestions for the manuscript.

Reviewer 3 Report

Line 19: "improved by 4.4884%" in comparison to what specific procedure/method?

Line 155: Unexplained abbreviation "PSO-WLSSVR" is used.

Equations used in the paper should be checked (after the export to PDF), in some cases they include apparently wrong symbols/characters (e.g. "M" or "L" in Equations 2, 3, respectively). 

Line 352: What are "BA and FA algorithms", unexplained abbreviations are used here, citations of source literature for both algorithms are missing.

What data was used to design the software sensor - simulation data (see text on Line 552) or experimental data (as seems to be indicated by the text on Lines 567-581)? 

Line 562: What is the relationship of the "nonlinear soft sensor model" mentioned here to the multi-model software sensor proposed in the paper?

Line 564: What was the specific form of the "complex nonlinear function" mentioned here?

Line 587: Does the nature of the offline measured data (especially the enzyme concentration) allow their interpolation between the measurement points?

How robust is the proposed software sensor (in online prediction mode) in relation to the variability in the length of the individual process phases? For example, in a situation where the length of the initial "adjustment" phase is significantly shorter or longer than the lengths captured in the training data set.

Round 2

Reviewer 1 Report

It was clear that the authors had put in substantial effort to generate the results they have. The revised version has a lot of improvements in presenting the idea.

About the questions I raised earlier, the authors seem to have addressed all of them. I do have a few minor comments/suggestions for future research. These are not related to the main theme of the manuscript and hence I would leave it to the discretion of authors if there has to be any change in the present article. 

(i) Measurement of protein yield in liters: I am not really convinced that protein yield is measured in liters, unless we are talking about 'pure protein'. In the projects I worked on, we used to report it as the protein concentration -grams/liter. Maybe I missed something here.

(ii) The optimization problem in Equation 8: I would write this in the following way.
First I would show the entire problem given by Equation 8, with all decision variables. Then, I would explain that this is a very complex problem and is therefore being decomposed in to few smaller steps. In each step, we would obtain the value for the one or more of the decision variables. Finally when S, n, omega are all known, I would formulate the problem again, but now with only theta being the decision variable and show that this is indeed an MIQP that is much easier to solve.
Again, this is a minor comment and I would let the author decide what is the best way to convey what they have done.

(iii) In the present work, I believe the ARX parameters (theta) are estimated following the minimization of the one-step prediction error, which is perfect considering that the identified models were able to give good predictions. However, estimating the parameters based on infinite-step prediction error may potentially result in a better ARX model. Even though this estimation problem is more complex, I would expect the particle swarm optimization algorithm to be efficient in finding a solution to the problem in reasonable time. I would suggest the authors to consider this in a future research problem.

(iv) While the first two parts of Question 6 were clearly answered, I am not entirely convinced about the third part - the rationale behind choosing the metaheuristic over a gradient based solver for solving a convex problem. But this may be ignored for now considering that the the algorithm used here could indeed identify a good set of a parameters for the model.

Author Response

This manuscript is a resubmission of an earlier submission. The following is a list of the peer review reports and author responses from that submission.

Round 1

Reviewer 1 Report

The authors propose a multi-model soft sensor modeling method based on improved compression factor (ICF), particle swarm optimization (PSO) and piecewise affine (PWA) to estimate the key biomass parameters in Pichia pastoris fermentation process in real time.

The article need major changes.

1. Introduction
1) Line 27, Line 32, ... : Correct the way to make the bibliographic citation throughout the text!

2) Line 48: The phrase 'The soft sensor technology is powerful, cost-effective, and compatible.' is incomplete. What is the soft sensor 'compatible' with? 

3) Lines 48, 49 and 50: The subject in the sentence 'Besides, using the divide and conquer modeling strategy, building a multi-site soft sensor model can greatly simplify the model structure, save calculation time, and fully mine the information of each sample.' needs to have a counterpoint before it, in order for the sentence to make sense. In short, why is a 'multi-local soft sensor model' necessary? Why 'save calculation time' is important ? 
   Which means the expression 'fully mine the information of each sample.' ?

4) Line 56: Could not find reference 'W.G. Zhang, K. Cai, Soft sensor modeling of alkaline protease fermentation process based on improved LWPLS, Sens. Mi- 559
crosyst. 39(2020)108-110+114.'.

5) Put a reference to 'Therefore, this paper improves the PSO with an improved compression factor (ICF).'.

2. Materials and Methods 
2.1. Description of modeling algorithm 
6) Lines 91 and 448: There is a mistake on f(.)

7) Equations 3, 10 and 11: The letter 'L' appears improperly in the equation.

8) Line 111: There is a mistake on f^(.,.).

9) Equation 7: There is a typo next to the equation numbering.

2.3.1. ICF-PSO algorithm
10) Lines 256 and 257: There is no information in the text about how f2(x), f4(x) and f5(x) are multi-objective problems.

11) Line 257: It's necessary to identify f2(x), f4(x) and f5(x) on Table 1.

12) Lines 261 and 262: This sentence 'The four above are excellent swarm intelligence algorithms that converge quickly.' it's about ICF-PSO, PSO, BA and FA. If these four algorithms converge quickly, why was ICF-PSO developed?

13) Figure 3: In order to assert that a swarm-type algorithm performs better, converges faster, and is more robust, it is necessary that tests include several experiments of the algorithms used for the same initial population and also some different initial populations. A good setup is: use 03 different random initial populations and for each of these populations run the algorithms 30 times.

2.5. Modeling steps of the PWA soft sensor model
14) Lines 417 and 418: It is essential to show how the ICF-PSO algorithm is used to optimize the multi-objective optimization problem (Equation 9). The multi-objective optimization problem has issues that do not exist in mono-objective, for example, 'How to maintain a diverse population in order to prevent premature convergence and achieve a well distributed, wide spread trade-off front? This has an impact on the Pareto front.'
    There is some information in these publications:
    - Diversity Metrics in Multi-objective Optimization: Review and Perspective (https://ieeexplore.ieee.org/document/4290378?denied=)
    - A new mechanism for maintaining diversity of Pareto archive in multi-objective optimization (https://www.sciencedirect.com/science/article/abs/pii/S0965997810000451)
    - On Multi-Objective Evolutionary Algorithms (https://repositorio-aberto.up.pt/bitstream/10216/70394/2/50192.pdf)

2.6. Introduction of experimental work
15) Line 466: Change 'male' to 'make'.

16) Line 468 and 469: What exactly does this phrase say 'Through coordinate transformation and interpolation operation on 10 batches of sample data'? What was the transformation and how was the interpolation done?

3. Results and Discussion
17) Some information is missing:
   - How many local models does the soft sensor have?
   - what are the parameters of each of the local models?

18) It is necessary to show the convergence of the ICF-PSO to optimize the two cost functions.

Reviewer 2 Report

The paper is in general poorly written, the scientific language is not precise, the design choices are not well justified with respect to the existing literature, a state-of-the art is not reported. It is not clear if the focus of the paper is on the application or on the optimization algorithm. The paper should be re-organized according to this choice.

Some examples of the encountered problemes are reported in the following (the list is not exaustive):

-The paper title is too long and contains typos.

-The last sentence of the abstract is not precise, with respect to which model do you obtained an improvement? In general from it is not possible to understand why a so complex procedure is needed and which are the benefit with respect to other SS design techniques (e.g. nonlinear ones).

As regards the introduction:

-what is an expression system? an explaination is needed for this Journal's audience. Why the advantages are 'obvious'? Is this relevant? On line 27,34 etc... I see some numbers? why? are they reference? in this case use the proper style.

-In line 46 you probably use the term 'parameters' instead of 'variables', as you surely know parameters and variables are not the same. Also you confuse measures with estimations, please use a rigorous scientific language. -The reasons for which you are using a multi-local model are not clear at all. -You should add a rigorous discussion of the state-of-the art in SS research and the contextualize and justify the method you are proposing.

-'The existing sample data evidently testify the nonlinearity of Pichia pastoris fermentation': why? how can the reader understand this 'evidence'? which sample data? Also you state that measuring the output variable is difficult and errors can frequently occur, so, which data are you using to develop the SS? what about data quality?

-formulas are poorly written, also k is not a 'time series' but the discrete time value;

-formulas have a double numeration...check a paper before submitting it!

-quality of fig. 1 is poor;

-the SS design procedure lacks of many details and makes it difficult to appreciate the results;

-the number of samples is very low with respect to the parameters to be optimized;

-a comparison with another method would improve the paper;

Round 2

Reviewer 1 Report

The reviewers' recommendations were all considered.